# Optimization of Cutting Parameters to Minimize Wall Deformation in Micro-Milling of Thin-Wall Geometries

**DOI:** 10.3390/mi16030310

**Published:** 2025-03-06

**Authors:** Ahmet Hasçelik, Kubilay Aslantas, Bekir Yalçın

**Affiliations:** 1Iscehisar Vocational High School, Afyon Kocatepe University, 03200 Afyon, Turkey; ahascelik@aku.edu.tr; 2Department of Mechanical Engineering, Faculty of Technology, Afyon Kocatepe University, 03200 Afyon, Turkey; bekiryalcin@aku.edu.tr

**Keywords:** micro-milling, micro thin wall, Taguchi and ANOVA analysis

## Abstract

Thin-walled micro-structures are a critical component of micro-devices, and their precise manufacture has a direct impact on product performance. Micro-milling is an effective manufacturing method that enables the production of micro-thin-walled structures with high precision and performance. Wall deformation is an undesirable problem in the production of parts with complex geometries and high aspect ratios, particularly when the height-to-thickness ratio (h/t) exceeds 20. In the micro-milling process, cutting parameters are the main factors affecting wall deformation. Therefore, optimising the cutting parameters is critical for the accuracy and precision of the cutting process. In this study, thin walls of 50 µm thickness, 1 mm height and 10 mm length were machined from an Al6061-T6 alloy using a tungsten carbide cutting tool with a diameter of 1 mm. The effects of feed rate, spindle speed and depth of cut cutting parameters (control parameters) used in the micro-milling process on the cutting forces and wall deformation (outputs) were investigated. A Taguchi L18 orthogonal design was used to optimise the cutting parameters. During the micro-milling experiments, the cutting forces were recorded, and the amount of deformation occurring in the thin wall was accurately determined using an optical profilometer with a motorised measuring system. Taguchi and ANOVA analyses were performed on the measured values of F_x_ tangential force, F_y_ feed force and thin-wall deformation to determine the effect of the control parameters on the outputs and to determine the most suitable cutting parameters to minimise deformation and keep the cutting forces under control. As a result of this study, the cutting parameter with the highest effect on the tangential force F_x_ was the depth of cut, with 56.94%, while the most effective cutting parameter on the feed force F_y_ was the feed rate, with 45.3%. The most effective parameter on the machined thin-wall deformation was the feed rate, with 87.36%. This study on the optimisation of cutting parameters in micro-thin-walled structures covers a unique topic that has been addressed in limited numbers in the literature.

## 1. Introduction

Mechanical and structural components where the wall thickness (t) is less than the wall height (h) are defined as thin-walled structures [1]. For a component to be defined as a thin-walled structure, the h/t ratio must be greater than 10 [2]. Components with a wall thickness of less than 100 µm are defined as micro-thin-walled structures. These delicate structures, which are part of micro-scale devices, are used in aerospace and aircraft components [3], micro fuel cells and chips [4], micro-scale heat exchangers [5], and micro-optical and electronic devices [6,7].

Micro-milling is one of the most commonly used methods to produce thin-walled micro-structures [8,9]. Micro-milling allows thin-walled parts to be produced accurately at high cutting speeds. Kou et al. [10] developed a novel method to reduce deformation in micro-thin-walled structures and successfully milled thin-walled structures with a thickness of 15 µm. Hasçelik and Aslantas [11] found that wall deformation increased with an increasing feed rate in thin-wall milling. Due to the low stiffness of thin-walled parts, they tend to deform easily during micro-milling [12]. This deformation can affect the process accuracy and lead to a deterioration in both the wall geometry and the machined surface quality [13]. There are many different factors that cause plastic deformation during the micro-milling of thin-walled structures [14]. The choice of cutting parameters (cutting speed, feed per tooth, axial and radial depth of cut) has a major influence on thin-wall deformation [15]. The feed rate is known to be more effective than other parameters [16]. Therefore, it is important to select cutting parameters in the optimum range in terms of minimum wall deformation [17]. However, problems arising from the size effect in micro-milling should not be ignored [18]. When the uncut chip thickness is less than the minimum chip thickness, ploughing occurs due to the negative rake angle effect. This situation can cause both an increase in cutting forces [19] and a deterioration in the surface quality of the workpiece [20].

Statistical experimental design methods, of which the Taguchi method is an example, are widely used in manufacturing processes to optimise cutting parameters. The Taguchi method is particularly effective in reducing the number of experiments required while still providing robust results [21]. It uses orthogonal arrays to systematically vary parameters and determine their effects on output variables. Utilising techniques such as signal-to-noise (S/N) ratio and analysis of variance (ANOVA), the Taguchi method assists in determining the optimal gauge settings to minimise variability and enhance process stability. This method has been successfully employed in various micro-milling studies to optimise parameters, such as surface roughness, tool wear and energy consumption. The Taguchi method utilises an R^2^ ratio, defined as the percentage of variation, to assess the model’s accuracy. It is widely acknowledged that a high R^2^ ratio corresponds to a superior predictive capability of the model [22]. The Taguchi method is frequently employed in micro-milling processes to optimise both the machining life and surface roughness [23]. Researchers have conducted studies to optimise various processes in micro-milling operations, with the aim of enhancing workpiece surface quality. These studies have focused on reducing surface roughness [24] and cutting tool wear [25], vibration [26], and energy consumption [27,28]. Additionally, the use of optimisation studies has enabled the detection and prevention of deformation in the machining of thin-walled structures [29,30]. It is noteworthy that the existing literature contains a paucity of studies focusing on optimisation for micro-thin-walled structures.

In this study, a comprehensive investigation was conducted to ascertain the impact of cutting parameters on cutting forces and wall deformation in the fabrication of thin-wall geometries by micro-milling. The cutting parameters were optimised using the Taguchi method and ANOVA analysis to achieve minimum cutting forces and wall deformation, without ignoring the ploughing effect in the micro-milling process.

## 2. Materials and Methods

### 2.1. Workpiece Material and Cutting Tool

Aluminium alloys are frequently preferred in industrial applications due to their heat and electrical conductivity, resistance to chemical effects, high rigidity and corrosion resistance [31]. These alloys facilitate structural efficiency in thin-walled structures, a property attributable to their light weight and ease of machining [32]. The T6 heat treatment applied to the Al6061 alloy has been shown to enhance its mechanical properties, though it should be noted that its machinability becomes somewhat more challenging as a result [33]. Al6061 is extensively employed in aerospace for thin-walled structures, particularly in less critical components, where a balance of strength, corrosion resistance and cost is imperative. Its reliability is well documented, and it is considered an essential material in the aerospace, electronics and automotive industries [34]. For these reasons, the Al6061-T6 alloy was selected as the workpiece material in this study. However, in order to minimise the wear effects that may occur in the tool, it was desired to use a material that can be machined easily enough (compared to alloys such as titanium).

In the present study, tungsten carbide cutting tools with a titanium-dipped silicon nitride (TiSiN) coating were utilised for the purpose of cutting. These tools are composed of a tungsten carbide alloy with a cobalt content of 90% WC and 10% Co. The diameter of the cutting tool was selected to be 1 mm ± 6 µm, which is approximately 20-times larger than the thin-wall thickness (50 µm) to minimise the elastic deformation of the tool (Figure 1a). The geometrical properties of the cutting tool are given in the table in Figure 1. It should be noted that there may be discrepancies in the micron-scale diameter measurements of the tools. It is acknowledged that such discrepancies can have a direct impact on the wall thickness of thin-walled geometries during the milling process. These diameter differences were considered during the cutting tests. The edge radius of the cutting tool is a significant geometrical parameter in micro-milling. The edge radius of the cutting tools used was determined by using SEM images. In the measurements, the minimum edge radius was 2.67 µm, while the maximum edge radius was determined as 2.93 µm. Accordingly, the edge radius of the cutting tool was determined to be 2.8 µm on average in SEM analyses (Figure 1c).

### 2.2. Taguchi Optimisation Draft

It has been established that the cutting parameters directly impact the deformation of the wall during micro-milling of thin-walled structures. The feed rate is identified as one of the most effective factors among the cutting parameters [16]. Consequently, meticulous attention was devoted to the feed rate during the experimental design phase, with the objective of conducting a comprehensive investigation into the ploughing and size effects, including the minimum chip thickness. The selection of cutting parameters has been shown to directly impact not only the deformation of the cutting wall but also the stability of the cutting forces and the efficiency of the process. In this context, the cutting tool manufacturer’s recommendations were taken into consideration in the selection of cutting parameters. In consideration of the studies that were examined in Section 1, the experimental study encompassed six distinct feed rates (f_z_), three varying rotational speeds (n), and three different axial depths of cut (a_p_), as delineated in Table 1. Given that this study encompasses full slot milling, it is notable that the radial depth of cut (a_e_) is equivalent to the tool diameter. The combinations of these parameters were planned with an L18 orthogonal array using the Taguchi mixed-level method (Table 2). The results of the experiments were analysed with an approach that aimed not only to optimise the cutting forces but also to minimise the wall deformation. This methodological approach ensured the attainment of optimal outcomes in a high-precision process such as micro-milling. To enhance the comprehension of the impact of cutting parameters on wall deformation, advanced analytical methodologies, such as signal-to-noise ratio (S/N), analysis of variance (ANOVA), regression modelling analysis and result validation, were employed. This comprehensive approach has enabled the meticulous evaluation of the individual parameters’ contributions to the process, thereby significantly enriching the existing literature on the subject.

### 2.3. Experimental Setup

The micro-milling experiments were conducted using a specially designed high-speed machining setup, which was tailored for precision and stability (Figure 2a). This setup features an IMT spindle capable of operating at a maximum speed of 60,000 rpm, ensuring high rotational accuracy. All components of the system were mounted on a vibration-free optical table. This was essential for eliminating potential instabilities during machining, which is crucial for achieving the desired thin-wall geometries. To ensure precise axial and radial movements, the setup employs linear guides (Thorlabs) with a resolution of 0.1 µm, allowing for accurate positional adjustments. The movement of these guides was controlled via dedicated computer software, enabling seamless execution of micro-milling operations. During the experiments, cutting forces were measured using a Kistler (Kistler City, Winterthur, Switzerland) (9119AA1) (v2.6.3) miniature dynamometer. Securely attached to the workpiece holder, the dynamometer captured real-time force signals, which were processed using the Dynoware software (v3.3.2.0), converting the raw data into measurable cutting force values. The data acquisition frequency employed for the measurement of cutting forces is 12.5 kHz. This frequency value is approximately ten-times higher than the tooth passing frequency at the highest speed employed in this study (35,000 rpm). The cutting forces (F_x_ and F_y_) measured in the cutting experiments are of particular significance as critical output parameters of the micro-milling process. The F_x_ force is representative of the tangential force, whilst the F_y_ force is representative of the feed force. It has been established that an increase in feed rate and depth of cut results in an increase in chip load on the tool, which consequently leads to an increase in cutting forces. Conversely, attaining elevated cutting speeds through augmented revolution numbers results in a reduction in cutting forces. The introduction of the ploughing effect is, thus, a causative factor in the increase in cutting forces. Consequently, the present study incorporated the alteration in cutting forces due to varying feed rates within the optimisation studies (Figure 3). Peak-to-valley amplitude, a common parameter in micro-milling studies, was used to analyse cutting forces, capturing the range between the maximum and minimum values. The resulting cutting force measurements were determined by averaging a minimum of three separate measurements. To facilitate visual observation of the cutting process, a USB microscope (Dino-Lite AM4113ZTL) (v1.5.29) was employed, mounted on a flexible stand (Dino-Lite RK-06A) (AnMo Electronics Corporation, New Taipei City, Taiwan) to allow for ease of movement and focus adjustment during experimentation. This arrangement enabled close monitoring of cutting tool behaviour and workpiece deformation. As illustrated schematically in Figure 2b, the micro-thin-wall milling process is performed in a specific manner. In order to measure the wall deformation after milling, a thin wall was formed only on the top and bottom surfaces of the specimen. The subsequent section will provide a comprehensive explanation of the methodology employed to measure the deformation of the micro thin wall.

### 2.4. Deformation Measurement in Micro Thin Wall

Subsequent to the execution of the cutting operations, a high-precision optical surface profilometer (Nanovea 3D ST400) (v5.2.9) (AnMo Electronics Corporation, New Taipei City, Taiwan) was utilised to detect the deformation of the micro thin wall (Figure 4). The aforementioned profilometer is based on the principle of focusing white light on the workpiece through an optical lens. Initially, the position of the reference point (x = 0 and y = 0) on the x and z axes of the thin wall to be measured was determined. The position of the optics was then fixed 50 µm from the top surface of the thin wall. The profilometer is equipped with a motorised table system, enabling precise movement of the sample along the x-axis at a rate of 3 mm/s for the entire length of the wall (10 mm). This process was repeated at 150 µm intervals, with a total of six measurements taken along the height of the wall. The measurement results were analysed with the MountainsMap software (v6.2.0.6746) of Digital Surf. The analysis of the measurement results is based on the highest deformation value. This measurement technique has not been previously documented in the literature and facilitates the acquisition of data along the entire length and height of the thin wall. The measurement results were then utilised to optimise the cutting parameters, ensuring a minimum deformation range.

## 3. Results and Discussion

### 3.1. Signal-to-Noise (S/N) Ratio and ANOVA Analyses

The analysis of the cutting parameters and their effects on output variables, such as tangential force (F_x_), feed force (F_y_) and deformation of the machined thin walls, was carried out using the Taguchi method and analysis of variance (ANOVA) techniques. The utilisation of Minitab software (version 22) was instrumental in facilitating these evaluations. The experimental design and parameter combinations summarised in Table 1 and Table 2 formed the basis for these analyses. The Taguchi analysis of variance was then used to determine the effect of the cutting parameters on the recorded forces (Table 3 and Table 4). The statistical significance of each parameter was determined using the ANOVA approach. The F-value, a measure of the variance ratio, is indicative of the degree of parameter influence. In this context, an F-value exceeding 4 is indicative of statistical significance. Additionally, the *p*-value serves to quantify the strength of the observed effect; values closer to 0 indicate stronger effects.

Taguchi analysis of variance revealed that the axial depth of cut (a_p_) emerged as the most significant parameter affecting the F_x_ force, accounting for 56.944% of the variation. This finding is consistent with the proposition that increasing the axial depth of cut increases the chip load to be cut by the cutting tool and, thus, increases the cutting forces. Conversely, spindle speed (n) was found to be the least contributing parameter to F_x_, with a contribution rate of 5.589%. Conversely, the feed rate (f_z_) demonstrated a statistically significant effect on F_x_, with a contribution rate of 34.02%. For the F_y_ force, feed rate emerged as the most influential parameter, contributing 45.201% to the observed variation. Axial depth of cut followed closely with a contribution of 42.432%, while the effect of spindle speed was minimal at 8.423%. The findings of this study demonstrate the predominant impact of feed rate and axial depth of cut on both F_x_ and F_y_ forces.

Subsequent S/N ratio analysis further clarified the optimum levels for the cutting parameters. The optimum configuration for F_x_ was determined as 0.1 µm/z feed rate, 35,000 rpm spindle speed and 50 µm axial depth of cut. These values correspond to the lowest cutting forces, as demonstrated in Table 5 and Figure 5. A similar outcome was observed in the F_y_ force analysis, which identified a 35,000 rpm spindle speed and 50 µm axial depth of cut as the optimum conditions, albeit with a slightly higher optimum feed rate of 0.25 µm/z (Table 6 and Figure 6). At feed rates less than 0.75 µm/z, irregularities appear due to the ploughing effect, which increases the cutting forces [20,35,36]. This phenomenon is particularly evident at a feed rate of 0.5 µm/z, where both cutting forces are quite high (Figure 7). At feed rates higher than 0.75 µm/z, an increase in cutting forces is observed due to increased chip load. The ploughing effect causes the cutting tool to compress and deform the material surface instead of forming chips. The ploughing effect leads to irregularities in the cutting forces because the cutting edge performs sliding and frictional movements on the material instead of chip formation. This process leads to increased surface roughness and the deterioration of surface quality. While the optimum feed rate for F_x_ was determined as 0.1 µm/z, this value was determined as 0.25 µm/z for F_y_. This can be interpreted as indicating that the low chip load at 0.1 µm/z and 0.25 µm/z feed rates is dominated by the ploughing effect.

The analysis of variance demonstrated that the feed rate was the most effective parameter and contributed 87.362% to the deformation change (Table 7). At low feed rates, deformation of the workpiece is attributed to the ploughing effect, while at high feed rates, deformation increases with the increase in chip load, thereby revealing the significant effect of feed rate on wall deformation. This effect is further substantiated by the negligible *p*-value documented in Table 7. It was further established that the axial depth of cut, a dominant factor in cutting forces, did not exert a significant impact on wall deformation. Nevertheless, it is noteworthy that it ranks as the second most effective parameter in terms of wall deformation, surpassed only by the feed rate. The effect of the number of revolutions on wall deformation was determined to be at the lowest level. This minimal effect is further substantiated by the F value being less than 1, indicating that the impact of the number of revolutions on wall deformation is not statistically significant, and its effect on process performance is limited.

In analysing the experimental design data using the S/N ratio transformation in the Taguchi method, the ‘smaller is better’ quality criterion for deformation values was taken as a basis. This approach provides a suitable evaluation criterion when the objective is to minimise deformation. In order to enhance the stability of the process and minimise deformation, it is essential to optimise the S/N ratios, taking into account this quality characteristic. As demonstrated in Table 8 and Figure 7, the cutting parameters that resulted in the lowest thin-wall deformation were identified as f_z_ = 0.75 µm/z, n = 35,000 rpm and a_p_ = 50 µm. It is evident that an increase in the number of revolutions and a decrease in the depth of cut result in a substantial reduction in wall deformation. However, if the feed rate is reduced below 0.75 µm/z, instabilities emerge within the process due to the ploughing effect, which, in turn, leads to increased wall deformation. Conversely, an increase in the feed rate to 1 µm/z and above results in an elevated chip load on the cutting tool, thereby inducing an increase in wall deformation. It is notable that at a feed rate of 2 µm/z, the deformation reaches its maximum value.

During the micro-milling tests, the cutting process and the deformation of the workpiece were meticulously observed through the utilisation of a USB microscope. The numerical value of the deformation deviation in the thin wall was ascertained through measurement using an optical profilometer and is exhibited in the scanning electron microscope (SEM) image of the thin wall. The deformation of the thin-wall geometries that exhibited the best and worst results in terms of deformation was then compared (Figure 8). In the L18 experimental design, the optimal result was obtained from the cutting test performed at cutting parameters 4-2-1 (f_z_ = 0.75 µm/z, n = 25,000 rpm and a_p_ = 50 µm), while the poorest deformation result was obtained from the cutting test performed at cutting parameters 6-1-3 (f_z_ = 2 µm/z, n = 15,000 rpm and a_p_ = 200 µm).

### 3.2. Interaction Between Significant Parameters

The interaction between the cutting parameters is of paramount importance, and interaction graphs facilitate the comparison of the level change of one input parameter with the level change of another input parameter. These graphs assist in the visual determination of the input parameter level that provides the desired performance. In the event of the lines in the graph being non-parallel, this indicates an interaction between the factors, and, thus, the effect of one factor is contingent on the other factors. The x-axis of the figure corresponds to one of the three levels of each variable, with the remaining levels indicated by lines of various colours. This representation facilitates the observation of how the variables of interest, i.e., the feed rate, the depth of cut, and the speed, affect the results. Furthermore, it reveals the presence of significant interactions between these variables that may have an effect on the response. Figure 9, Figure 10 and Figure 11 present interaction plots as a function of the respective input parameters for F_x_, F_y_ and deformation, respectively.

The results demonstrate the impact of feed rate, axial depth of cut, and speed on process performance. Figure 9 and Figure 10 illustrate the intersecting lines of feed rate and depth of cut, which are observed to be parallel. This finding suggests that the variation in these parameters exerts an independent influence on cutting forces. In Figure 11, the lines are not parallel, indicating an interaction between the cutting parameters in terms of wall deformation.

The optimisation study that was conducted following the cutting tests revealed that changes in feed rate and axial depth of cut significantly affected the cutting forces, while the spindle speed had a relatively limited effect in comparison to these two parameters. With regard to thin-wall deformation, the feed rate emerged as the most critical parameter. The effect of the two most effective cutting parameters (feed rate and axial depth of cut) on the cutting forces F_x_, F_y_ (Figure 12 and Figure 13) and thin-wall deformation (Figure 14) was demonstrated using contour plots. The utilisation of these plots facilitates the determination of optimum parameter ranges by visualising the interaction of multiple response variables. The optimal parameter ranges determined through this method were consistent with the results obtained from the response tables. In general, the contour plots demonstrated that lower depths of cut and feed rates resulted in lower cutting forces and thin-wall deformation. However, due to the ploughing effect, it was observed that feed rates in the range of 0.6 to 0.8 µm/z provided superior results in comparison to those below 0.6 µm/z. This observation underscores the pivotal role of feed rate in influencing process stability and deformation in micro-milling operations.

### 3.3. Regression Modeling Analysis

First-order mathematical models were developed for thin-wall deformation (1), F_x_ tangential force (2) and F_y_ feed force (3) using Minitab software. Upon analysis of the linear regression results, the percentage of variation in R Square (R^2^), which is a measure of the model’s accuracy, was found to be 96.55% for F_x_, 96.06% for F_y_ and 92.19% for thin-wall deformation.(1)$$\mathrm{Processed}\;\mathrm{thin}\;\mathrm{wall}\;\mathrm{deformation}=16.09-5.62\;f_{z_{0.10}}-6.67\;f_{z_{0.25}}-2.82\;f_{z_{0.5}}-7.75\;f_{z_{0.75}}-7.24\;f_{z_{1}}+\;30.1\;f_{z_{2}}+1.72\;n_{15,000}+0.81\;n_{25,000}-2.53\;n_{35,000}-3.12\;a_{p_{50}}-0.14a_{p_{100}}+3.26\;a_{p_{200}}$$(2)$$F_{x}\;\mathrm{tangential}\;\mathrm{force}=1.1476-0.2886\;f_{z_{0.10}}-0.2199\;f_{z_{0.25}}+0.0331\;f_{z_{0.5}}-0.0619\;f_{z_{0.75}}+0.0334\;f_{z_{1}}+\;0.5041\;f_{z_{2}}+0.1321\;n_{15,000}-0.0111\;n_{25,000}-0.1209\;n_{35,000}-0.3868\;a_{p_{50}}-0.0341a_{p_{100}}+0.4209\;a_{p_{200}}$$(3)$$F_{y}\;\mathrm{feed}\;\mathrm{force}=0.8327-0.2127\;f_{z_{0.10}}-0.2393\;f_{z_{0.25}}+0.0817\;f_{z_{0.5}}-0.1930\;f_{z_{0.75}}+0.1917\;f_{z_{1}}+\;0.3717\;f_{z_{2}}+0.0890\;n_{15,000}+0.0505\;n_{25,000}-0.1395\;n_{35,000}-0.2418\;a_{p_{50}}-0.0566a_{p_{100}}+0.2985\;a_{p_{200}}$$

As illustrated in Figure 15, the predicted values for cutting forces and thin-wall deformation are presented as a function of the actual values. These graphs provide important visual evidence of the predictive ability of the model. The normal probability plot was utilised to assess the model’s accuracy and appropriateness. The arrangement of the data along a straight line in the plot is indicative of the model’s precise representation of the relationship between the predicted and actual values. This analysis, thus, demonstrates the statistical reliability and satisfactory accuracy of the model’s predictions with regard to cutting forces and thin-wall deformation.

### 3.4. Optimization Results’ Verification

Subsequent to the optimisation of the cutting parameters by means of the Taguchi approach, the lowest cutting forces and thin-wall deformation were obtained at 35,000 rpm and 50 µm axial depth of cut. While the optimum speed and depth of cut parameters are consistent across all three output parameters, the feed rate parameter exhibits variation. The optimum feed rate was determined to be 0.1 µm/z for F_x_, 0.25 µm/z for F_y_ and 0.75 µm/z for thin-wall deformation. It is notable that these three combinations do not feature among the cutting parameters in the L18 experimental design. Consequently, verification experiments were conducted utilising the optimal parameters of f_z_ = 0.1 µm/z, n = 35,000 rpm, a_p_ = 50 µm for F_x_, f_z_ = 0.25 µm/z, n = 35,000 rpm, a_p_ = 50 µm for F_y_ and f_z_ = 0.75 µm/z, n = 35,000 rpm, a_p_ = 50 µm for thin-wall deformation (Table 9). The results were then compared with the data estimated using the Taguchi method (Figure 16). The outcomes of the validation experiments and the Taguchi prediction results demonstrate a mean acceptable error of 23.11%.

## 4. Conclusions

The objective of this study was to optimise the cutting parameters for the micro-milling of Al6061-T6 thin-walled workpieces with tungsten carbide cutting tools, utilising the Taguchi approach. The experimental phase of this study involved the execution of micro-milling tests utilising the Taguchi L18 orthogonal test sequence. The cutting forces, F_x_ and F_y_, were meticulously recorded during the course of these tests. Subsequent to the cutting tests, wall deformation was measured on the surface of the thin-walled workpieces. Subsequent to the collection of all results, they were analysed using both the Taguchi and ANOVA approaches. This process led to the identification of the following findings.

The Taguchi approach, utilised for the optimisation of the cutting parameters, yielded highly accurate results and demonstrated a high level of performance. The effects of cutting parameters on tangential force (F_x_), feed force (F_y_) and thin-wall deformation were analysed with a confidence level of 96.55%, 96.06%, 92.19% (R-sq), respectively.It is evident that a decrease in the number of revolutions and an increase in the feed rate and depth of cut will result in an increase in cutting forces and thin-wall deformation.The findings of this study indicated that the most effective parameter on F_x_ tangential cutting force was depth of cut, with a percentage of 56.94%, and that the most effective parameter on F_y_ feed force was the feed rate, with a percentage of 45.3%. In the case of machined thin-wall deformation, the most effective parameter was identified as the feed rate, with an impact factor of 87.36%.The impact of depth of cut and machine speed on the micro-milled thin-wall deformation was found to be limited to 3.26% and 1.58%, respectively.In analyses of variance (ANOVA) determining the effect of control parameters on the output parameters, high residual plot values of 92.19%, 96.55% and 96.06% were obtained for micro-milled wall deformation, F_x_ tangential force and F_y_ feed force, respectively.It was determined that the ploughing mechanism exhibited enhanced efficacy in inducing thin-wall deformation through micro-milling when compared to conventional cutting forces. The predominant factor contributing to this phenomenon is hypothesised to be the substantial influence of the feed rate on thin-wall deformation.The optimal cutting parameters for micro-milled thin-wall deformation were determined as f_z_ = 0.75 µm/z, n = 35,000 rpm, a_p_ = 50 µm. Taguchi analysis estimated these with an average error of 19.44%.First-order mathematical models for micro-milled thin-wall deformation (1) and cutting forces (2) and (3) were developed using regression analyses.Optimised cutting parameters minimised deformation in the micro-milling of thin-walled structures. These results are seen as a guide for precision manufacturing in the aerospace, electronics and medical device industries.

## Figures and Tables

**Figure 1 micromachines-16-00310-f001:**
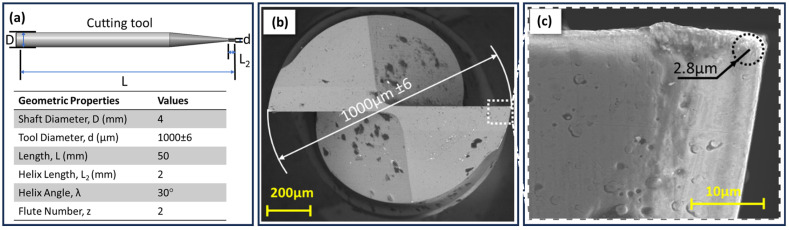
(**a**) Geometric values of the cutting tool, (**b**) SEM images of tool diameter, (**c**) tool edge radius.

**Figure 2 micromachines-16-00310-f002:**
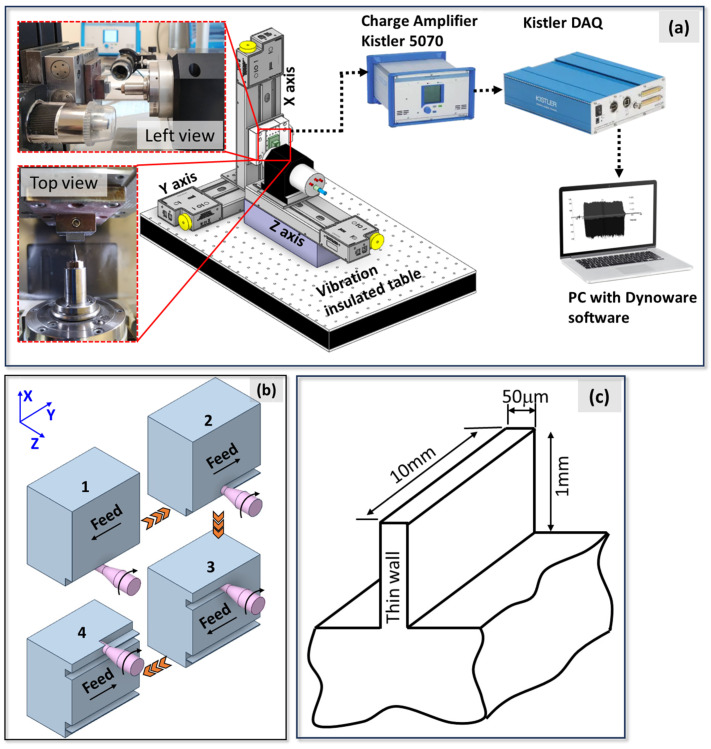
(**a**) Overview of the experimental setup, (**b**) the cutting process for thin-wall geometry, (**c**) micro-thin-wall dimensions.

**Figure 3 micromachines-16-00310-f003:**
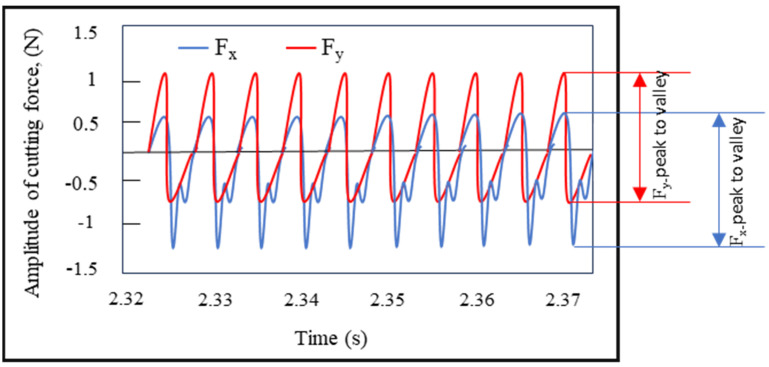
Example of cutting force signals for peak-to-valley F_x_ and F_y_ force data.

**Figure 4 micromachines-16-00310-f004:**
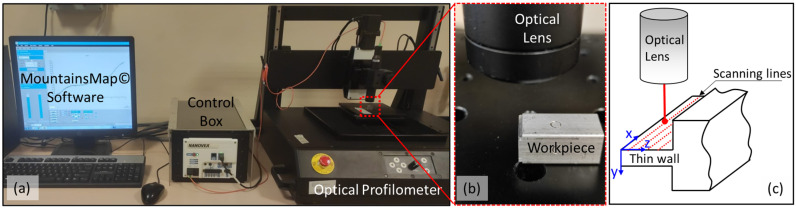
(**a**) Optical profilometer device, (**b**) thin-wall deformation scanning, (**c**) scanning method on the thin-wall surface.

**Figure 5 micromachines-16-00310-f005:**
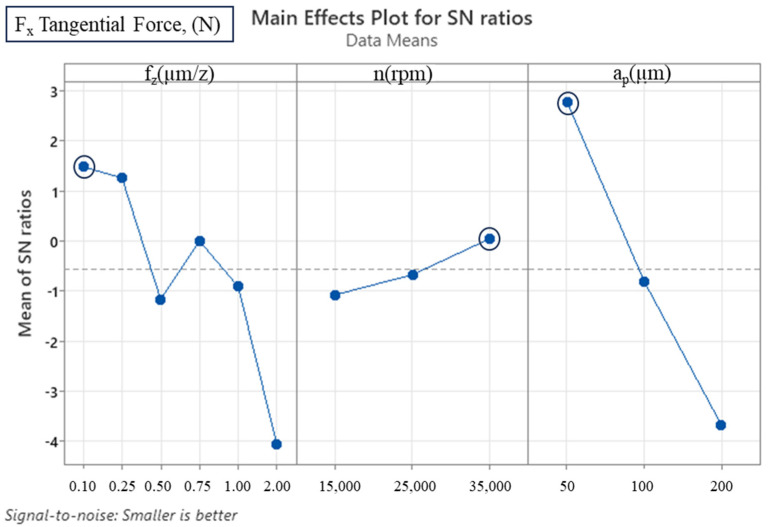
S/N ratio plot for F_x_ tangential force (smaller is better).

**Figure 6 micromachines-16-00310-f006:**
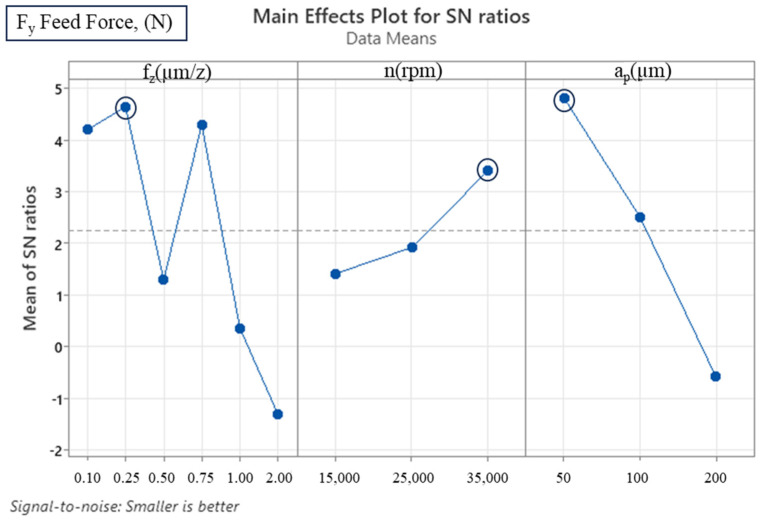
S/N ratio plot for F_y_ feed force (smaller is better).

**Figure 7 micromachines-16-00310-f007:**
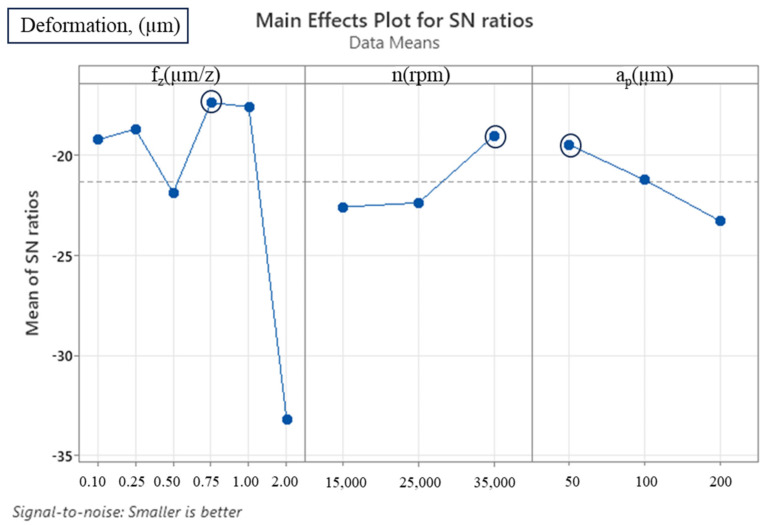
S/N ratio plot for thin-wall deformation (smaller is better).

**Figure 8 micromachines-16-00310-f008:**
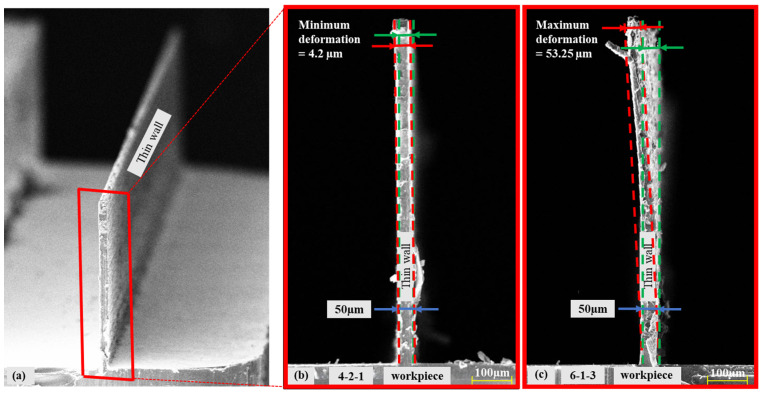
(**a**) SEM image of thin-wall geometry, thin-wall geometries with the (**b**) lowest and (**c**) highest levels of deformation. (Green dashed lines represent undeformed thin wall geometry. Red dashed lines represent the deformed thin wall geometry).

**Figure 9 micromachines-16-00310-f009:**
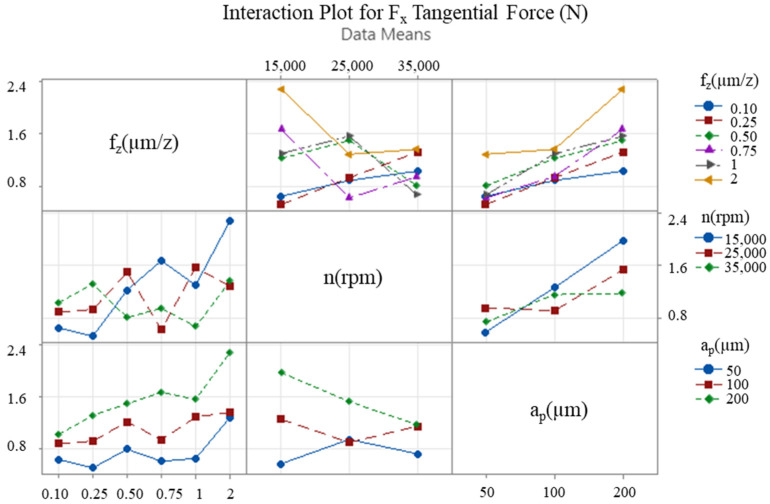
Interaction plot for F_x_ tangential force.

**Figure 10 micromachines-16-00310-f010:**
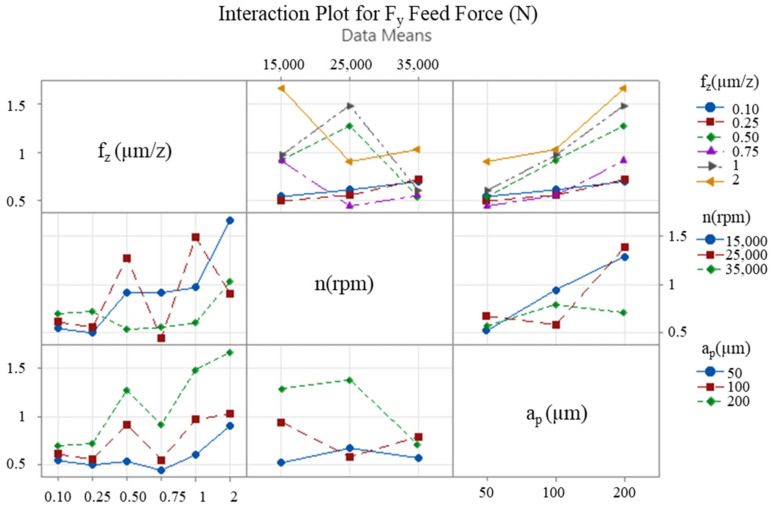
Interaction plot for F_y_ feed force.

**Figure 11 micromachines-16-00310-f011:**
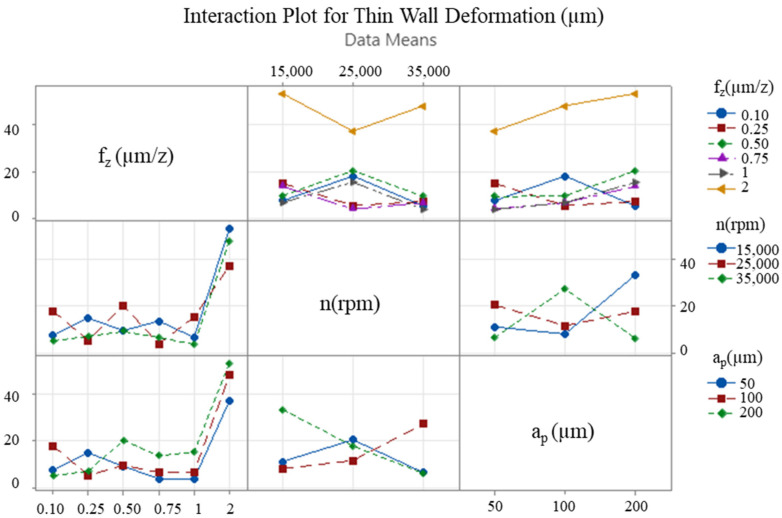
Interaction plot for thin-wall deformation.

**Figure 12 micromachines-16-00310-f012:**
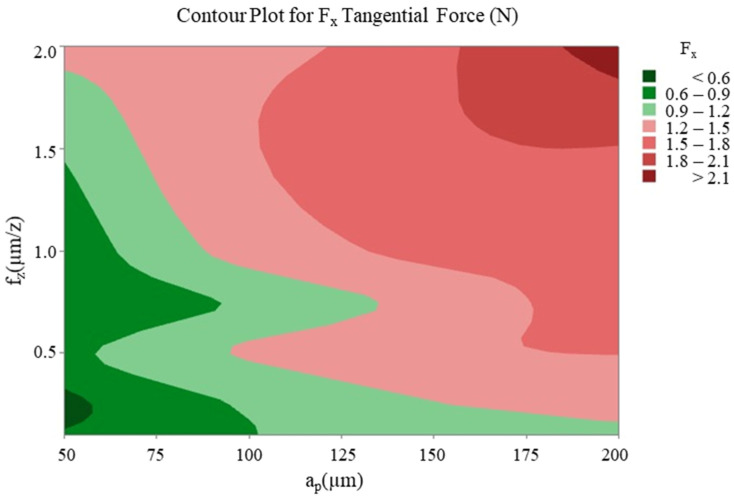
Contour plot of feed rate and axial depth of cut for F_x_ tangential force.

**Figure 13 micromachines-16-00310-f013:**
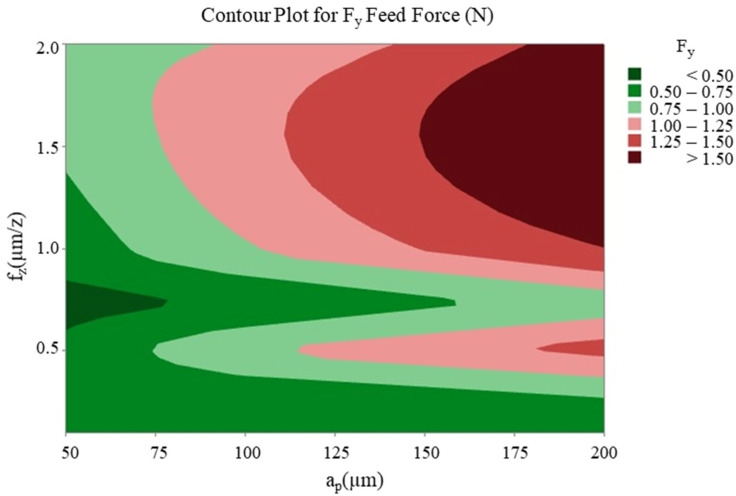
Contour plot of feed rate and axial depth of cut for F_y_ feed force.

**Figure 14 micromachines-16-00310-f014:**
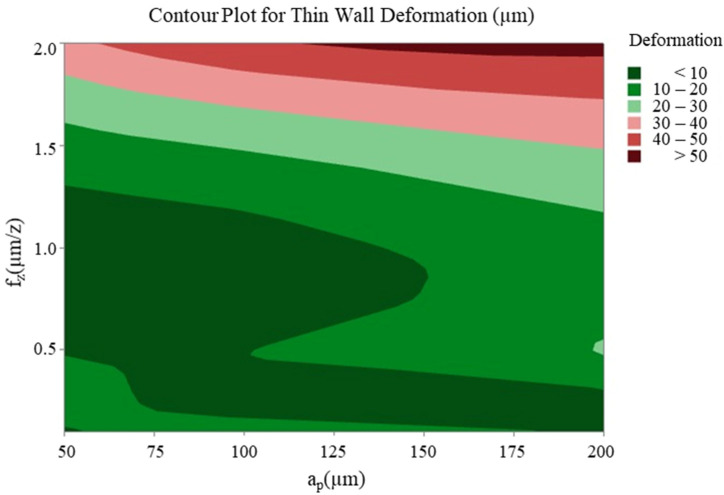
Contour plot of feed rate and axial depth of cut for thin-wall deformation.

**Figure 15 micromachines-16-00310-f015:**
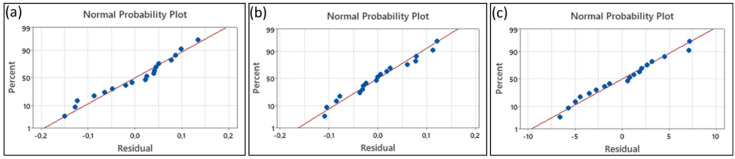
Residual plots for (**a**) F_x_ tangential force, (**b**) F_y_ feed force, (**c**) thin-wall deformation.

**Figure 16 micromachines-16-00310-f016:**
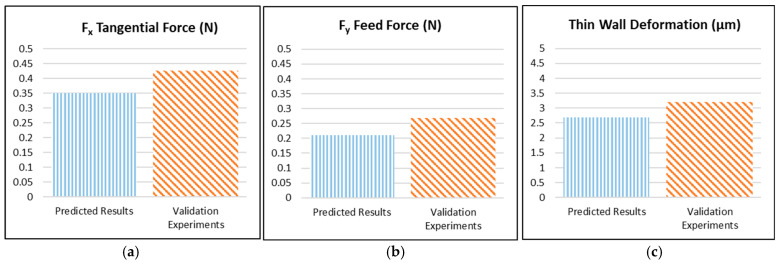
Results of validation experiments (**a**) F_x_ tangential force, (**b**) F_y_ feed force, (**c**) thin-wall deformation.

**Table 1 micromachines-16-00310-t001:** Cutting parameters used in micro-milling tests.

Control Factors			Levels			
	1	2	3	4	5	6
Feed Rate, f_z_ (µm/z)	0.1	0.25	0.5	0.75	1	2
Spindle Speed, n (rpm) Cutting Speed, V_c_ (m/min)	15,000 (47.1)	25,000 (78.5)	35,000 (109.9)			
Depth of cut, a_p_ (µm)	50	100	200			

**Table 2 micromachines-16-00310-t002:** Taguchi experiment design with L18 orthogonal array.

Control Factors	L18 Orthogonal Experimental Design																	
	1	2	3	4	5	6	7	8	9	10	11	12	13	14	15	16	17	18
Feed rate (f_z_)	1	1	1	2	2	2	3	3	3	4	4	4	5	5	5	6	6	6
Spindle speed (n)	1	2	3	1	2	3	1	2	3	1	2	3	1	2	3	1	2	3
Depth of cut (a_p_)	1	2	3	1	2	3	2	3	1	3	1	2	2	3	1	3	1	2

**Table 3 micromachines-16-00310-t003:** Analysis of variance for F_x_ tangential force in the micro-milling for thin wall.

Source	DF	Contribution	Adj SS	Adj MS	F-Value	*p*-Value
Feed rate (µm/z)	5	34.020%	1.1754	0.23507	15.79	0.001
Spindle speed (rpm)	2	5.589%	0.1931	0.09657	6.49	0.021
Depth of cut (µm)	2	56.944%	1.9674	0.98372	66.09	0.000
Error	8	3.447%	0.1191	0.01488		
Total	17	100%	3.4550			

**Table 4 micromachines-16-00310-t004:** Analysis of variance for F_y_ feed force in the micro-milling for thin wall.

Source	DF	Contribution	Adj SS	Adj MS	F-Value	*p*-Value
Feed rate (µm/z)	5	45.201%	0.96380	0.19276	18.34	0.00034
Spindle speed (rpm)	2	8.423%	0.17961	0.08980	8.54	0.01034
Depth of cut (µm)	2	42.432%	0.90476	0.45238	43.03	0.00005
Error	8	3.944%	0.08410	0.01051		
Total	17	100%	2.13226			

**Table 5 micromachines-16-00310-t005:** Response table for F_x_ tangential force signal-to-noise ratios.

Level	1	2	3	4	5	6	Delta	Rank
Feed rate (µm/z)	1.48020	1.25830	1.16376	0.00275	0.90571	4.05702	5.53722	2
Spindle speed (rpm)	1.07151	0.67022	0.04912				1.12063	3
Depth of cut (µm)	2.76866	0.80309	3.65818				6.42684	1

**Table 6 micromachines-16-00310-t006:** Response table for F_y_ feed force signal-to-noise ratios.

Level	1	2	3	4	5	6	Delta	Rank
Feed rate (µm/z)	4.1981	4.6374	1.3044	4.2915	0.3653	−1.3039	5.9413	1
Spindle speed (rpm)	1.4169	1.9249	3.4046				1.9876	3
Depth of cut (µm)	4.8085	2.5112	0.5732				5.3817	2

**Table 7 micromachines-16-00310-t007:** Analysis of variance for thin-wall deformation in the micro-milling.

Source	DF	Contribution	Adj SS	Adj MS	F-Value	*p*-Value
Feed rate (µm/z)	5	%87.362	3307.14	661.43	17.89	0.0004
Spindle speed (rpm)	2	%1.588	60.11	30.06	0.81	0.4771
Depth of cut (µm)	2	%3.236	122.49	61.25	1.66	0.2501
Error	8	%7.814	295.8	36.98		
Total	17	%100	3785.56			

**Table 8 micromachines-16-00310-t008:** Response table for thin-wall deformation signal-to-noise ratios.

Level	1	2	3	4	5	6	Delta	Rank
Feed rate (µm/z)	−19.24	−18.7	−21.87	−17.38	−17.6	−33.2	15.82	1
Spindle speed (rpm)	−22.58	−22.37	−19.04					3
Depth of cut (µm)	−19.48	−21.24	−23.28					2

**Table 9 micromachines-16-00310-t009:** Comparison of prediction results with experimental results and error rates.

Output Parameters	Parameter Setting Level	Predicted Results	Validation Experiments	Error %
F_x_ tangential force (N)	f_z_ = 0.1 µm/z, n = 35,000 rpm, a_p_ = 50 µm	0.35	0.428	22.28
F_y_ feed force (N)	f_z_ = 0.25 µm/z, n = 35,000 rpm, a_p_ = 50 µm	0.21	0.268	27.62
Thin-wall deformation (µm)	f_z_ = 0.75 µm/z, n = 35,000 rpm, a_p_ = 50 µm	2.69	3.213	19.44

## Data Availability

The data is contained in this paper.

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
