# Peer review of "Optimization of Cutting Parameters to Minimize Wall Deformation in Micro-Milling of Thin-Wall Geometries"

_micromachines, 2025, doi:10.3390/mi16030310_

Round 1
Reviewer 1 Report
Comments and Suggestions for Authors
Paper details an experimental study for milling of micro thin walled structures. Taguchi techniques are used to reduce experimental testing. Forces and wall deformation were measured.
Clarification in the abstract to the size of the walls and geometries could be provided. IT is quite generic.
I would like to see cutting speed quoted rather than spindle speed.
Is the definition of thin walled structure correct? I would not detail a wall height of 1.1 mm compared to wall thickness of 1mm as think walled yet it would be given the definition. A quick search online finds that “Thin-walled structures are defined by their geometric property where the wall thickness is significantly smaller than other dimensions, such as length and width” and Thin-Walled Assumption: The assumption that the wall thickness is small enough to neglect shear deformation across the thickness. A discussion on this could help. Thin-walled structures journal states diameter to thickness ration is D/t is greater than 20. You are probably machining thin walled structures but the detail could be clearer.
It would be good to provide further details of studies 13 and 14.There could be numbers in this section as well for the literature review.
I think the authors need to be careful on the statements of using Taguchi method. As it is factorial design, then difficult to optimise. Clarification of this would help.
|What was the error on the tool edge radius measurements. It would be good to provide details on this.
How was the dynamic frequency of the dynamometer taken into account when measuring the forces.
The cutting process could be clarified. What was the wall dimensions, etc.
It would be interesting to know the affect of tool wear. Was this significant?
It would be good to see the interactions plots for the results. To see if there were interactions. These cant be easily calculated (more tests needed which would defeat objects but the plot would indicate if an issue.
Why is their a dip in values for 0.5um/z feedb rate.
Higher magnification images of the deformation would be beneficial. / Figure 8 (b) is difficult to tell. The caption needs improving. Also error for the measurement method would be good. Detailed as being 53.25um. Can your system measure to 0.01um? More images of this would be good too.
Ranges for the predicted and validation experiments could be provided.
Error on the results ranged from 19.44 to 27.62. I would not define this as highly accurate.
Reference list could be in full unless required. i.e. et al. better to detail all authors out of respect.
Good number of references but more could be brought out in the results and discussion.
Author Response
Manuscript Number: micromachines-3485393
Optimization of cutting parameters to minimize wall deformation in micro-milling of thin wall geometries
First of all, we would like to thank the editor and all three referees for their valuable comments and contributions. We would like to express that the manuscript is more quality and understandable in line with the comments and suggestions.
RC: Reviewer Comment AA: Author Answer
Reviewer #1:
RC1- Paper details an experimental study for milling of micro thin walled structures. Taguchi techniques are used to reduce experimental testing. Forces and wall deformation were measured.
AA1-Thank you so much
RC2- Clarification in the abstract to the size of the walls and geometries could be provided. IT is quite generic
AA2- Additional clarifications on thin wall dimensions were given in the Abstract. Thank you for your attention
RC3- I would like to see cutting speed quoted rather than spindle speed.
AA3- We absolutely agree with the referee on this issue. However, in many studies on micromachining, spindle speed is used instead of cutting speed. Therefore, spindle speed was preferred. If the referee agrees, we ask that it be kept as spindle speed. Thank you for your understanding.
RC4- Is the definition of thin walled structure correct? I would not detail a wall height of 1.1 mm compared to wall thickness of 1mm as think walled yet it would be given the definition. A quick search online finds that “Thin-walled structures are defined by their geometric property where the wall thickness is significantly smaller than other dimensions, such as length and width” and Thin-Walled Assumption: The assumption that the wall thickness is small enough to neglect shear deformation across the thickness. A discussion on this could help. Thin-walled structures journal states diameter to thickness ration is D/t is greater than 20. You are probably machining thin walled structures but the detail could be clearer.
AA4- In this regard, the referee is absolutely correct. In other words, for a geometry to be a thin wall, the ratio of height to thickness (h/t) must be at least 20. In this study, it is stated on page 3 that the h/t ratio is 20. In this study, the wall height is 1mm and the wall thickness is 50microns. An explanatory schematic expression has been added to Figure 2.
RC5- It would be good to provide further details of studies 13 and 14.There could be numbers in this section as well for the literature review.
AA5- Studies 13 and 14 are presented in more detail. Thank you.
RC6- I think the authors need to be careful on the statements of using Taguchi method. As it is factorial design, then difficult to optimise. Clarification of this would help.
Traditional factorial design of experiments can be very costly and time consuming and therefore researchers prefer to avoid parameter selection with full factorial experiments. Therefore, Taguchi method, which can determine the effect of different milling parameters on cutting force and other outputs more economically and scientifically, is preferred. In this study, it was used as a tool to determine or optimize (select the best) the effect of each parameter on the result by correlating the effect of different milling conditions on milling forces and milled thin wall deformation. According to the best results, the equation related to the outputs was developed. In many studies in the literature, the term optimization is used in studies with parameters determined by Taguchi method. Thank you very much for your contributions.
(https://www.sciencedirect.com/search?qs=taguchi)
RC7- What was the error on the tool edge radius measurements. It would be good to provide details on this
AA7- This explanation is given on page 3 and the following statement has been added to the manuscript. Thank you.
“The edge radius of the cutting tools used was determined by using SEM images. In the measurements, the minimum edge radius was 2.67 µm, while the maximum edge radius was determined as 2.93 µm.”
RC8- How was the dynamic frequency of the dynamometer taken into account when measuring the forces.
AA8- The detail about the data acquisition frequency is given on page 5 and the following statement has been added to the manuscript. Thank you.
The data acquisition frequency employed for the measurement of cutting forces is 12.5 kHz. This frequency value is approximately ten times higher than the tooth passing frequency at the highest speed employed in the study (35,000 rpm).
RC9- The cutting process could be clarified. What was the wall dimensions, etc.
AA9- Micro thin wall dimensions are given in figure 2
RC10- It would be interesting to know the affect of tool wear. Was this significant?
AA10- In micro milling, the edge radii increase with tool wear. This leads to an increase in cutting forces, which is expected to increase the thin wall deformation. In this study, a new tool was used in each experiment and tool wear was not considered. This suggestion of the referee also led us to a different study to be conducted in the future. Thank you
RC11- It would be good to see the interactions plots for the results. To see if there were interactions. These cant be easily calculated (more tests needed which would defeat objects but the plot would indicate if an issue.
AA11- With the Taguchi method, it is possible to produce multiple plots to show the effect of input parameters on outputs: linear, areal, two-dimensional, etc. In this study, two-dimensional surface plots were preferred instead of interaction plots. The reason for this is to show the effect of inputs on outputs more strikingly on an areal plane. It is thought that the relationships can be seen more clearly with spatially colored two-dimensional graphs. Thank you for your advice.
RC12- Why is their a dip in values for 0.5um/z feedb rate.
AA12- In micro-milling operations, a ‘ploughing’ effect occurs at feed values smaller than the minimum chip thickness. This causes the cutting tool to compress and deform the material surface instead of forming chips. The ploughing effect leads to irregularities in the cutting forces because the cutting edge performs sliding and frictional movements on the material instead of chip formation. This process leads to increased surface roughness and deterioration of surface quality. Please see studies given below
1- Weule, H., Hüntrup, V., & Tritschler, H. (2001). Micro-cutting of steel to meet new requirements in miniaturization. CIRP Annals, 50(1), 61-64.
2- Vogler, M. P., DeVor, R. E., & Kapoor, S. G. (2004). On the modeling and analysis of machining performance in micro-endmilling, Part I: Surface generation. Journal of Manufacturing Science and Engineering, 126(4), 685-694.
The following statement has been added to section 3.1.
“At feed rates less than 0.75 µm/z, irregularities appear due to the ploughing effect which increases the cutting forces [20, 35]. This phenomenon is particularly evident at a feed rate of 0.5 µm/z, where both cutting forces are quite high (Figure 7). At feed rates higher than 0.75 µm/z, an increase in cutting forces is observed due to increased chip load. Ploughing effect causes the cutting tool to compress and deform the material surface instead of forming chips. The ploughing effect leads to irregularities in the cutting forces because the cutting edge performs sliding and frictional movements on the material instead of chip formation. This process leads to increased surface roughness and deterioration of surface quality. While the optimum feed rate for Fx was determined as 0.1 µm/z, this value was determined as 0.25 µm/z for Fy. This can be interpreted as indicating that the low chip load at 0.1 µm/z and 0.25 µm/z feed rates is dominated by the ploughing effect.”
RC13- Higher magnification images of the deformation would be beneficial. / Figure 8 (b) is difficult to tell. The caption needs improving. Also error for the measurement method would be good. Detailed as being 53.25um. Can your system measure to 0.01um? More images of this would be good too.
AA13- A general view of the micro thin wall is attached to Figure 8. The measurement method is detailed in section 2.4. The Nanovea optical profilometer used in deformation measurements can measure with an accuracy of 0.01 microns. You can find details in the link below.
https://nanovea.com/profilometers/
RC14- Ranges for the predicted and validation experiments could be provided.
AA14-The predicted values and the results of the verification experiment are given in Table 10 and the error rate between these results is calculated mathematically and shown in the same table.
RC15- Error on the results ranged from 19.44 to 27.62. I would not define this as highly accurate.
AA15-highly accurate is used for R2 ratios. For error rates between predicted values and validation experiments, the term acceptable has been added to the last sentence of section 3.3.
RC16- Reference list could be in full unless required. i.e. et al. better to detail all authors out of respect.
AA16- According to the editorial rules of this journal, et al. is used for names with more than three authors. Authors do not have a preference in this regard.
RC17- Good number of references but more could be brought out in the results and discussion.
AA17- Additional references were used in the results and discussion section

Reviewer 2 Report
Comments and Suggestions for Authors
The authors conducted research on optimization of cutting parameters to minimize wall deformation in micro-milling of thin wall geometries. The manuscript is well organized and the experimental methods are appropriately designed. After addressing the following issues, it is recommended to accept.
1. In the "Introduction" section, the introduction to the Taguchi experiment can be simplified.
2. The basis for selecting each data in Table 1 needs to be explained.
3. Is the test result based on the average of multiple tests?
4. The experimental results reflected in Figure 8 require some theoretical explanation.
5. It is suggested to add some microstructure images of the processed surface in the paper to facilitate the explanation of the experimental results.
6.In the conclusion, it is suggested to provide guidance on engineering applications.
Author Response
Manuscript Number: micromachines-3485393
Optimization of cutting parameters to minimize wall deformation in micro-milling of thin wall geometries
First of all, we would like to thank the editor and all three referees for their valuable comments and contributions. We would like to express that the manuscript is more quality and understandable in line with the comments and suggestions.
RC: Reviewer Comment AA: Author Answer
Reviewer #2:
The authors conducted research on optimization of cutting parameters to minimize wall deformation in micro-milling of thin wall geometries. The manuscript is well organized and the experimental methods are appropriately designed. After addressing the following issues, it is recommended to accept.
RC1- In the "Introduction" section, the introduction to the Taguchi experiment can be simplified.
AA1- The Taguchi section in the “Introduction” was simplified.
RC2- The basis for selecting each data in Table 1 needs to be explained.
AA2-The feed rate parameter was selected more than the number of spindle speed and depth of cut parameters. Thus, it is aimed to observe the effect of the scraping mechanism. Similar studies in the literature for the cutting tool and workpiece pair were taken as reference for the selection of numerical values. AA2 Section 2.2 explains why the parameters were chosen. The present description is elaborated. Thank you
RC3- Is the test result based on the average of multiple tests?
AA3- Test results differ for cutting forces and deformation. The following sentences have been added to sections 2.3 and 2.4. Thank you for your advice
“The resulting cutting force measurements were determined by averaging a minimum of three separate measurements.”
“The analysis of the measurement results is based on the highest deformation value.”
RC4- The experimental results reflected in Figure 8 require some theoretical explanation.
AA4- How the deformation measurements in Figure 8 are performed is explained in section 2.4. The explanation referring to Figure 8 and Figure 8 have been revised.
RC5- It is suggested to add some microstructure images of the processed surface in the paper to facilitate the explanation of the experimental results.
AA5- The SEM image of the processed surface is added to Figure 8a.
RC6- In the conclusion, it is suggested to provide guidance on engineering applications.
AA6- The following sentence has been added as the last sentence of the conclusion. Thank you for your advice.
“Optimised cutting parameters have minimised deformation in the micro-milling of thin-walled structures. These results are seen as a guide for precision manufacturing in the aerospace, electronics and medical device industries”

Round 2
Reviewer 1 Report
Comments and Suggestions for Authors
- I still do not like the comments regarding Taguchi and optimisation. If you use an array and dont asses interactions then you cannot optimise.
- The error calculations for the cutting edge are still not given but ok.
- Cutting speed should be given. The comment that other studies use spindle speed so that must be ok is poor. Just because other studies don't detail correctly, it does not mean that other papers should follow this.
- I was more interested in the resonant frequency of the dyno than the ratio of cutting force frequency measurement to tooth passing frequency.
- Figure 2c is better.
- I would be surprised if the measurement equipment can measure at that level detailed.
- Id still like to see interaction plots to see if there any.